# Risk of Chronic Disease after an Episode of Marasmus, Kwashiorkor or Mixed–Type Severe Acute Malnutrition in the Democratic Republic of Congo: The Lwiro Follow-Up Study

**DOI:** 10.3390/nu14122465

**Published:** 2022-06-14

**Authors:** Pacifique Mwene-Batu, Ghislain Bisimwa, Philippe Donnen, Jocelyne Bisimwa, Christian Tshongo, Michelle Dramaix, Michel P. Hermans, André Briend

**Affiliations:** 1Ecole Régionale de Santé Publique, Université Catholique de Bukavu, Bukavu P.O. Box 285, Democratic Republic of the Congo; ghislainbiba@yahoo.fr (G.B.); jocelynebintu@gmail.com (J.B.); 2Faculté de Médecine, Université de Kaziba, Kaziba P.O. Box 85, Democratic Republic of the Congo; 3Hôpital Provincial General de Reference de Bukavu, Université Catholique de Bukavu, Bukavu P.O. Box 162, Democratic Republic of the Congo; christshongo@gmail.com; 4Ecole de Santé Publique, Université Libre de Bruxelles, 1070 Brussels, Belgium; philippe.donnen@ulb.be (P.D.); michele.dramaix@ulb.be (M.D.); 5Division of Endocrinology & Nutrition, Cliniques Universitaires St-Luc, Université Catholique de Louvain, 1200 Brussels, Belgium; michel.hermans@saintluc.uclouvain.be; 6Department of Nutrition, Exercise and Sports, Faculty of Science, University of Copenhagen, DK-2200 Copenhagen, Denmark; andre.briend@gmail.com; 7Center for Child Health Research, Faculty of Medicine and Health Technology, Tampere University, 33520 Tampere, Finland

**Keywords:** chronic disease, acute malnutrition, marasmus, kwashiorkor, long-term effect, DR Congo

## Abstract

Background: Long-term impact of different forms of severe acute malnutrition (SAM) in childhood on the emergence of noncommunicable diseases (NCDs) is poorly known. Aim: To explore the association between subtypes of SAM during childhood, NCDs, and cardiovascular risk factors (CVRFs) in young adults 11 to 30 years after post-SAM nutritional rehabilitation. Methods: In this follow-up study, we investigated 524 adults (mean age 22 years) treated for SAM during childhood in eastern Democratic Republic of the Congo (DRC) between 1988 and 2007. Among them, 142 had a history of marasmus, 175 of kwashiorkor, and 207 had mixed-form SAM. These participants were compared to 407 aged- and sex-matched control adults living in the same community without a history of SAM. Our outcomes of interest were cardiometabolic risk markers for NCDs. Logistic and linear regressions models were sued to estimate the association between subtype of SAM in childhood and risk of NCDs. Results: Compared to unexposed, former mixed-type SAM participants had a higher adjusted ORs of metabolic syndrome [2.68 (1.18; 8.07)], central obesity [1.89 (1.11; 3.21)] and low HDL-C (High-density lipoprotein cholesterol) [1.52 (1.08; 2.62)]. However, there was no difference between groups in terms of diabetes, high blood pressure, elevated LDL-C (low-density lipoprotein cholesterol) and hyper TG (hypertriglyceridemia) and overweightness. Former mixed-type SAM participants had higher mean fasting glucose [3.38 mg/dL (0.92; 7.7)], reduced muscle strength [−3.47 kg (−5.82; −1.11)] and smaller hip circumference [−2.27 cm (−4.24; −0.31)] compared to non-exposed. Regardless of subtypes, SAM-exposed participants had higher HbA1c than unexposed (*p* < 0.001). Those with a history of kwashiorkor had cardiometabolic and nutritional parameters almost superimposable to those of unexposed. Conclusion: The association between childhood SAM, prevalence of NCDs and their CVRFs in adulthood varies according to SAM subtypes, those with mixed form being most at risk. Multicenter studies on larger cohorts of older participants are needed to elucidate the impact of SAM subtypes on NCDs risk.

## 1. Introduction

Severe acute malnutrition (SAM) is a major public health problem in low-income countries (LICs) [1]. SAM has two clinical presentations: marasmus, characterized by extreme weight loss with muscle and adipose tissue wasting, and kwashiorkor with nutritional edema localized in lower limbs or sometimes generalized. There is also a mixed form that includes both extreme weight loss and nutritional edema [1].

According to the developmental origin’s theory of noncommunicable diseases (NCDs), malnutrition in childhood predisposes people to NCDs in adulthood [2]. Despite growing evidence on the negative long-term effects of childhood undernutrition in high- and middle-income countries (HMICs), data related to the long-term outcomes of children treated for SAM in LICs are scarce [3,4,5,6,7].

Studies conducted in Uganda showed that recovery from wasting was associated with slightly increased diastolic blood pressure (BP) in adolescence, while participants who remained emaciated or stunted throughout the follow-up period had slightly lower BPs than adolescents with normal growth [8]. In Malawi, pre-pubescent survivors of childhood SAM had a pattern of thrifty growth (suggesting that growth of the torso and head was preserved to the detriment of that of the limbs) which is associated with greater risk of NCDs, even though their lipid profile, glucose tolerance, glycated hemoglobin A1c (HbA1c), salivary cortisol, and sitting height were not different from controls up to seven years after nutritional rehabilitation [9]. In addition, in a recent cohort study (Lwiro Cohort) from the Democratic Republic of Congo (DRC), SAM in childhood was associated with a higher risk of NCDs in adulthood, mainly as abnormal glucose homeostasis, metabolic syndrome (MetS) and visceral obesity 11 to 30 years after nutritional rehabilitation [10].

However, most of these studies did not take into account subtypes of malnutrition to which the participants were exposed, even though the risk of NCDs may differ depending on the SAM subtypes [7]. In a narrative review examining the evidence for differences in cardiovascular risk factors (CVRFs) between survivors of marasmus and kwashiorkor, it was observed that adults with a history of marasmus more frequently had low weight, stunted growth, greater risk of pancreatic beta-cell dysfunction/glucose intolerance, and non-alcoholic fatty liver than adults with a history of kwashiorkor [11].

This study, using secondary data of the Lwiro cohort study [10], aims at exploring the association between subtypes of SAM (marasmus, kwashiorkor and mixed-form) during childhood and NCDs and their cardiovascular risk factors (CVRF) in adulthood.

## 2. Methodology

### 2.1. Study Area

This study took place in a population living in the South Kivu, DRC, where people have monotonous, undiversified, and low-quality diet, and nutritional transition has not yet taken place [12].

### 2.2. Study Design and Population

This is an observational follow-up study comparing young adults with a history of previous hospital admission for SAM with community controls. The study was conducted among young adults who were treated for SAM during childhood at Lwiro pediatric hospital (LPH) between 1988 and 2007, still living in Miti-Murhesa and Katana HZ in 2018 [10,13]. A total of 1981 children were treated for SAM at LPH in the period of interest [13]. The nutritional status of the study participants at the time of their admission to hospital [10,14,15] was reassessed with the Emergency Nutrition Assessment (ENA) for SMART program, version October 2007, based on WHO child growth standards [16]. Based on these standards, 1664 children were classified as having SAM [13]. The remains were excluded from subsequent analyses. All children hospitalized for SAM were treated according to the guidelines used at that time [15].

For this study, 524 participants from the initial cohort who were still living in the two HZ were examined [13]. To assess long-term growth and health consequences of SAM, these survivors (SAM-exposed) were compared to 407 unexposed adult controls randomly-selected from the community [10,13,17].

Unexposed controls had no hospital history of SAM, were of the same sex, living in the same community, and less than 24 months older or younger than the exposed participants. Unexposed individuals were randomly selected by spinning a bottle at the exposed participant’s home and enquiring door to door, starting from the nearest house towards which the bottle pointed [10,13]. Though the optimal study design would be a 1:1 ratio of exposed and unexposed. However, unexposed participants proved harder to recruit than exposed participants, as many feared being associated with childhood SAM and its social stigma [10,13]. For that reason, a ratio of 0.75 non-exposed per exposed was eventually achieved [13,17].

The exposed were divided into three groups according to SAM subtypes during childhood, namely marasmus, kwashiorkor, or mixed SAM. The mixed-form, marasmus and Kwashiorkor was defined based on WHO child growth standards [16].

### 2.3. Data Collection

Because this is an additional analysis of an already published study, data collection (paper-based surveys) was described in detail elsewhere [10,13], so in this work we will only give the main features.

The questionnaire covered variables relating to the participant’s identity, their lifestyles (alcohol and tobacco consumption as well as dietary habits), their medical history, presence of known CVRFs (familial or personal), as well as SES [10,18].

During the follow-up, the anthropometric measurements considered were weight in Kg, height in cm, BMI [Body Mass Index (in kg/m^2^)], waist circumference and hip circumference in cm [10]. The anthropometric measurements were carried out in accordance with WHO guidelines [16] and were quality-controlled with two members of the team taking independent measurements [10,13]. Muscle strength in kilograms (kg) was measured with a Takei Grip-D device (Takei, Niigata, Japan) [10].

BP was measured using an electronic device (OMRON Hem 7001E^®^) [10]. Fasting glycemia was analyzed by a glucose oxidase method using a portable electronic glucose meter (Code free^®^) [10].

Lastly, 4 mL of blood was taken using an antecubital venipuncture after 12 h of fasting to determine the serum creatinine, Hb1Ac, lipid profile, and albumin using standard calorimetric enzymatic methods [10,19]. Quality control of blood analyses was performed using a lyophilized human serum by Cypress diagnostics [10]. Due to financial constraints, HbA1c analysis was done in a subgroup of 142 participants (90 exposed and 52 unexposed).

### 2.4. Outcomes

The main outcomes were NCDs, including diabetes mellitus (DM), hypertension (HBP), overall obesity, visceral and android obesity, metabolic syndrome (MetS), and dyslipidemia, assessed by their different routine clinical and biological markers [waist circumference, hip circumference, Waist to Height Ratio (WHtR) and Waist to Hip Ratio (WHR), Triglyceride (TG), total cholesterol, High density lipoprotein (HDL-C), Low Density Lipoprotein (LDL-C), glycated hemoglobin (HbA1c), fasting glycemia, albumin, creatinine and blood pressure (systolic, diastolic and mean)]. The definition of different NCDs was based on international standard [20,21,22,23,24,25,26,27]. We also assessed the nutritional status in adulthood.

Nutritional status in adulthood was assessed by anthropometry, both for the exposed and unexposed groups.

#### Potential Adult Confounder Variables

Other variables such as age, sex, socioeconomic status (SES), lifestyles (alcohol, tobacco and diet diversity), and family history of DM and/or HBP in first-degree relatives were added in the modeling as potential confounding factors.

SES was established based on a summative score taking into account the participant’s level of education and occupation, as well as their living conditions [10].

Food consumption frequency was evaluated using a food diversity score devised by the World Food Program [10,28,29]. For tobacco and alcohol exposure, the definition was based on international standard [30,31].

### 2.5. Statistical Analysis

We used Stata software, version 13.1. Categorical variables were summarized as frequency and proportion. Continuous variables were presented as mean and standard deviation (SD), or as median and range (min-max) depending on whether data distribution was symmetrical.

The data from exposed and unexposed were compared using Chi-squared tests or Fisher’s exact tests (for proportion) with Bonferroni’s correction.

Linear and logistic regression models were used, respectively, for the continuous variables [BMI, WC, HC, WHtR, WHR, muscle strength, lipid profile, HbA1c, fasting glycemia, albumin, creatinine and BP (systolic, diastolic and mean)] and dichotomous variables (overall obesity, thinness, visceral and android obesity, DM, HBP, MetS and dyslipidemia). For TG, we made a logarithmic transformation. The basic models only included the primary exposure subtype of SAM (No SAM, marasmus only, kwashiorkor only and mixed-type), giving a crude mean difference between exposed and unexposed for the quantitative variables, and crude odds ratios (ORs) for categorical variables. The mean differences and ORs are shown with 95% confidence intervals (95% CIs). For TG, the exponential of the regression coefficient provides the geometric means ratio.

Different models were then constructed to analyze the effects of SAM subtypes adjusted for dietary diversity score and SES. Potential confounders that did not differ substantially between groups were not considered. In addition, a multiple comparison with Bonferroni’s correction was added to each model. Conditions for applying linear and logistic regression were verified.

## 3. Results

### 3.1. Recruitment of Exposed Group

Based on WHO child growth standards, among 524 SAM-exposed participants, 142 (27.1%) were classified as marasmus, 175 (33.4%) as kwashiorkor, and 207 (39.5%) as mixed-form (Figure 1).

#### 3.1.1. Sociodemographic and Economic Characteristics of the Different Subgroups

Table 1 summarizes the sociodemographic and economic characteristics of the study population. The mean age was about 22 years in the different groups. No differences were observed in terms of sex distribution between different study groups.

Compared to unexposed, former mixed-type SAM and kwashiorkor participants had a less satisfactory food consumption, the differences being statistically significant. The composite indicator of SES, constructed from the variables of living conditions, education and occupation [10] tended to be higher in the unexposed than in SAM-exposed participants without reaching significance level.

Table 2 shows the prevalence of NCDs and their risk factors in the different subgroups. Former mixed-type SAM participants had significantly higher prevalence of MetS, low HDL-C and android obesity than unexposed ones. In addition, they had a higher prevalence of hypo-HDL-cholesterolemia compared to former kwashiorkor participants. On the other hand, there was no significant difference between the different groups regarding alcohol consumption, smoking and familial histories of HBP and/or DM.

Former mixed-form SAM participants had a higher risk of MetS, android obesity, and hypo-HDL-cholesterolemia than non-exposed after adjustment (Table 3). In addition, they had a greater risk of hypo-HDL-cholesterolemia compared with former kwashiorkor participants. However, no difference was observed between the different groups in terms of DM, HBP, high LDL-C and high TG, overweightness, and visceral obesity, even after adjustment.

#### 3.1.2. Mean Differences in Clinical and Biological Markers for NCDs between Subgroups

In terms of anthropometry, compared to unexposed, former mixed-type SAM participants had shorter height and lower weight. In addition, they had lower BMI and lower weight compared to participants with a history of kwashiorkor (Table 4). These differences were significant even after adjustment (Table 5). However, no difference was observed between former mixed-type SAM participants and those with a history of marasmus (Table 4 and Table 5). Finally, it was observed that former mixed-type SAM participants had a thinness prevalence 3 times higher than unexposed (Table 4). In addition, compared with unexposed, former mixed-type SAM participants had reduced muscle strength and smaller hip circumference, even after adjustment (Table 4 and Table 5).

Among the clinical and biological markers of NCDs, former mixed-type SAM participants had higher glycemia than SAM-unexposed (Table 4), a difference that remained significant after adjustment (Table 5). Former SAM participants, regardless of subtypes, had higher HbA1c than unexposed controls (Table 4 and Table 5).

Former mixed-type SAM participants had low total cholesterol, non-HDL-C and LDL-C compared to non-exposed (Table 4). However, this difference became nonsignificant after adjustment (Table 5). Former mixed-form SAM participants had lower HDL-C levels compared to those with a history of kwashiorkor, even after adjustment (Table 5). Finally, no difference was observed in TG levels between subgroups (Table 4 and Table 5).

Similarly, no difference was observed in BP (MBP, SBP, DBP and PP), creatinine and albumin between the different groups compared two by two (Table 4 and Table 5).

Finally, former marasmus participants had greater WHtR and WHR than unexposed controls, even after adjustment (Table 5).

## 4. Discussion

Our results suggest that, compared to unexposed controls, participants with a history of mixed-type SAM during childhood had the highest risk of developing NCDs as well as their CVRFs (MetS, android obesity and hypo–HDL-cholesterolemia) in adulthood, followed by those with a history of marasmus (as regards android obesity), whereas those who had kwashiorkor seem to have no long-term cardiometabolic sequelae. Our results also show that SAM in childhood exposes survivors to abnormal glucose homeostasis regardless of SAM-subtypes. To our knowledge, the present study is the first to evaluate in an LIC the association between different subtypes of SAM in childhood and NCDs and their CVRFs in adulthood after a long follow-up (between 11 and 30 years) in a context of endemic malnutrition.

We observed a higher prevalence of metabolic syndrome in former mixed-type SAM than unexposed participants, even after adjustment for SES and food consumption score. This may be linked to visceral obesity and insulin resistance. In addition, former mixed-type SAM participants had a more atherogenic lipid profile, characterized by low HDL-C, a hallmark of atherogenic dyslipidemia, associated to insulin resistance and hyperinsulinemia [10,24,25,26].

We also observed that compared to unexposed, a higher proportion of former mixed-type SAM and marasmus participants developed android obesity, with reduced hip circumference. This suggests an altered distribution of adipose tissue between visceral and gluteo-femoral depots, and/or a gluteo-sarcopenic component. Our findings are consistent with other studies which reported that SAM survivors are at significantly higher risk of visceral obesity [6,9,10]. Visceral fat is considered metabolically less favorable, particularly regarding the secretion of harmful adipokines. It is positively correlated with insulin resistance and chronic reactive hyperinsulinemia, which are linked to increased cardiometabolic risk, particularly atherosclerotic cardiovascular diseases [32,33,34].

Adults with a history of kwashiorkor in infancy had a prevalence of NCDs that was almost identical to that of unexposed controls. Our results corroborate those of other studies conducted in South Africa [35] and Uganda [36], which showed that survivors of childhood kwashiorkor did not show risk factors for NCDs 10 years after an episode of SAM and when they were 11–19 years old, respectively [35,36].

This difference could be explained by a higher frequency of intrauterine growth retardation (IUGR) and low birth weight (LBW) among children who suffered from marasmus during childhood [11,37]. When these risks factors are combined with SAM, they may have adverse effects that last into adulthood and put them at increased risk for excessive/ectopic fat deposition (12). In addition, LBW has been associated with an increased risk of developing insulin resistance, visceral adiposity, atherosclerosis and glucose intolerance [2,4,7]. The risk of NCDs becomes even greater when marasmus and kwashiorkor are combined, making participants with a history of mixed-type SAM at greater risk [38]. In this regard, the risk of morbi-mortality becomes significantly higher when multiple anthropometric deficits combine, compared to participants without deficits, with a dose-response association depending on the number of deficits [38]. In contrast, individuals with kwashiorkor were less likely to have experienced prenatal insults, as evidenced by adequate birth weight (BW) [11,37]. Thus, after recovery, they may have a return to normal metabolism, with less risk of developing NCDs [11,35,36].

With regard to glucose homeostasis, SAM-exposed participants, regardless of subtypes, had a slightly higher HbA1c than unexposed controls, even after adjustment: this would probably put them at higher risk of impaired glucose homeostasis in later life. These glucose abnormalities could be ascribed to inadequate nutrition in childhood, leading to decreased number/function of pancreatic β cells, in addition to acquired insulin resistance in malnutrition survivors, as a result of reduced fat free mass. Sarcopenia, which is associated with lower muscle glucose uptake, particularly in the postprandial period, could be another factor that may contribute to increased HbA1c levels [11]. However, this disorder of glucose metabolism was more pronounced in former mixed-type SAM participants, who also had higher glycemic averages than unexposed. This may reflect the presence of two deficits, including acquired sarcopenia, as the dose-response association depends on the number of deficits [38]. In contrast to unexposed, former mixed-type SAM participants had more prevalent thinness and reduced muscle strength, suggesting reduced fat free mass (FFM). Reduced FFM in adulthood is associated with decreased thermogenesis, insulin resistance/hyperinsulinemia, decreased fasting and postprandial glucose uptake, higher risk of MetS and/or type 2 DM, and increased incidence of atherothrombotic cardiovascular disease [6,7]. All of this could also explain to a large extent the higher prevalence of NCDs and their risk factors observed in former mixed-type SAM participants compared to other SAM subtypes.

Compared to unexposed, former mixed-type SAM participants had statistically significant lower weight and shorter standing height. This would suggest insufficient recovery, with persistent long-term effects of SAM on growth to adulthood, especially in those with mixed-form.

There are some limitations in our study. First, the major limitation is survival bias. In fact, only 524 of the 1981 participants of the initial cohort were examined, and they may have different characteristics from those of the unanalyzed. Nevertheless, we believe that this would not have significantly altered our main conclusions because the characteristics at hospital admission did not differ between the lost to follow-up and traced subjects 18. Second, the study design does not allow us to separate the detrimental effects of SAM per se from those of the child’s early environment or from persistently living in the same poor environment up to adulthood. Third, we did not have information on important risk factors, including, gestational age, BW and birth height, rate of growth in the first months of life, and growth between time of discharge from hospital and when the study was conducted. These variables could be potential confounding factors as they could be linked both with SAM and its potential negative long-term effects. Fourth, although not severely malnourished in the past, unexposed controls lived in the same unfavorable conditions as SAM-exposed participants, and it is difficult to establish whether they were perfectly healthy and well-nourished throughout the period of interest. In addition, we cannot entirely rule out a possible misdiagnosis of less severe malnutrition form, which probably did not lead them to consulting at the nutrition center in the area because they lived in the same unfavorable environment. This probably also justifies the fact that some differences were not observed between the two groups.

In conclusion, our results suggest that the long-term effects of childhood SAM vary according to SAM subtypes. Those with mixed-form are at the highest risk of subsequent NCDs and their risk factors, whereas those with a history of kwashiorkor appear not exposed to such risks. Multicentre studies involving larger cohorts of adults having recovered from SAM could provide greater understanding of the impact of SAM on the overall risk of CVDs and HBP in adulthood. Finally, our results should also remind us of the importance of fighting against SAM and its consequences as a public health priority.

## Figures and Tables

**Figure 1 nutrients-14-02465-f001:**
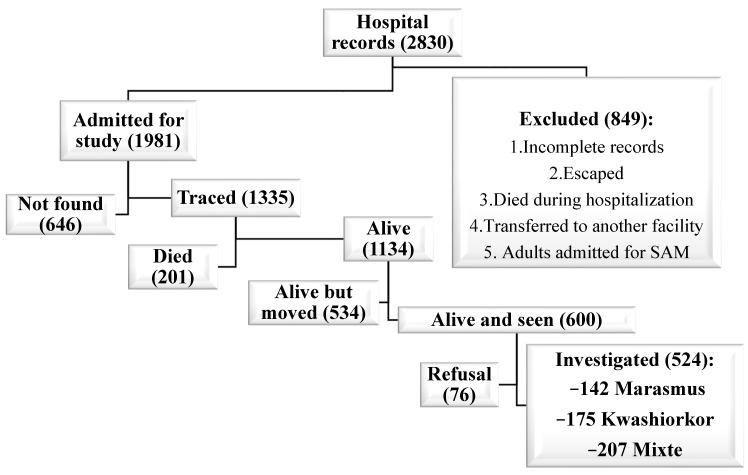
Recruitment of exposed group.

**Table 1 nutrients-14-02465-t001:** Sociodemographic and economic characteristics of the three SAM subgroups and unexposed controls.

	Marasmus		Kwashiorkor		Mixed-Type		Unexposed		*p* Value
	N	%	N	%	N	%	N	%	
**Age (years) Mean (SD)**	142	22.68 (4.63)	175	22.32 (4.03)	207	21.98 (4.63)	407	22.14 (4.62)	
**Male gender**		53.5		49.7		53.1		50.6	0.848
**Food consumption score**	142		175		207		407		
Insufficient		13.4		8.6		11.6		6.9	
Bordeline		37.3		42.3		38.2		31.7	0.011
Satisfactory		49.3		49.1 ^1^		50.2 ^1^		61.4	
**Socioeconomic status**	124		164		184		357		
Low		64.5		60.4		66.8		55.5	
Average		32.3		36.0		32.0		37.8	0.069
High		3.2		3.7		2.2		6.7	

^1^ Different from unexposed. *p* value calculated using the chi2 test (or Fisher’s exact when applicability conditions were required) Prevalence of NCDs and their risk factors in the different subgroups.

**Table 2 nutrients-14-02465-t002:** Unadjusted differences in cardiovascular risk factors (CVRF) and NCDs prevalence in the four groups.

	Marasmus		Kwashiorkor		Mixed-Type		Unexposed	
	N	%	N	%	N	%	N	%
**Dyslipidemia**	118		134		172		331	
High LDL-C		2.6		3.1		1.8		1.6
Low HDL-C		39.8		26.9^2^		46.5 ^1,2^		34.1
High TG		7.8		6.1		4.2		4.7
**Diabetes mellitus (DM)**	107	8.4	129	7.8	162	11.7	319	7.5
**Hypertension**	104	7.7	125	5.8	156	5.7	301	6.6
**Metabolic syndrome**	92	12.0	108	8.3	132	12.1 ^1^	265	4.9
**Overweight/Obesity**	141	12.8	167	18.0	201	9.0	396	13.1
**Visceral obesity**	136	52.9	161	54.0	195	50.8	372	43.8
**Android obesity**	142	61.3	171	64.9	203	72.4 ^1^	405	54.6
**Cardio-Vascular Risk factor**	142		175		207		407	
Alcohol (yes)		35.2		32.6		39.1		40.3
Tobacco (yes)		3.5		1.7		3.9		1.5
First-degree relative with HBP and/or DM (yes)		34.5		34.9		28.5		32.9

^1^ Significant difference from unexposed. ^2^ significant difference between mixed-type SAM and kwashiorkor. proportion (in each group) were compared using the chi2 test (or Fisher’s exact when applicability conditions were required). HDL-C = High-Density Lipoprotein Cholesterol. LDL-C = Low-Density Lipoprotein cholesterol. TG = Triglycerides.

**Table 3 nutrients-14-02465-t003:** Adjusted Odds Ratio of developing NCDs or their CVRFs (95% CIs) in SAM-exposed compared with unexposed.

Variable	Marasmus vs. Unexp	Kwash vs. Unexp	Mixed vs. Unexp	Kwash vs. Marasmus	Mixed-Form vs. Marasmus	Mixed-Type vs. Kawsh
**Overweight/Obesity**						
aOR (95% CI)	1.10 (0.50; 2.44)	1.39 (0.70; 2.76)	0.76 (0.35; 1.65)	1.27 (0.53; 3.03)	0.69 (0.27; 1.77)	0.54 (0.23–1; 0.28)
**Metabolic syndrome**						
aOR (95% CI)	2.38 (0.68; 8.24)	1.60 (0.45; 5.73)	2.68 (1.18; 8.07) ^1^	0.67 (0.17; 2.61)	1.13 (0.34; 3.71)	1.67 (0.49; 5.67)
**Hypertension**						
aOR (95% CI)	1.32 (0.40; 4.25)	0.74 (0.20; 2.67)	0.83 (0.26; 2.65)	0.56 (0.12; 2.49)	0.63 (0.15; 2.50)	1.12 (0.25; 4.90)
**Diabetes**						
aOR (95% CI)	0.84 (0.25; 2.77)	0.93 (0.32; 2.66)	1.34 (0.54; 3.29)	1.1 (0.28; 4.30)	1.59 (0.45; 5.50)	1.43 (0.47; 4.37)
**High TG**						
aOR (95% CI)	1.78 (0.55; 5.73)	1.25 (0.37; 4.14)	0.92 (0.26; 3.23)	0.69 (0.18; 2.66)	0.51 (0.13; 2.05)	0.74 (0.18; 3.03)
**Low HDL-C**						
aOR (95% CI)	1.31 (0.70; 2.44)	0.64 (0.34; 1.21)	1.52 (1.08; 2.62) ^1^	0.49 (0.23; 1.04)	1.16 (0.58; 2.29)	2.34 (1.18; 4.65) ^2^
**High LDL-C**						
aOR (95% CI)	2.05 (0.28; 15.08)	2.1 (0.34; 12.94)	0.92 (0.09; 8.84)	1.02 (0.12; 8.05)	0.45 (0.03; 5.18)	0.44 (0.04; 4.49)
**Visceral Obesity**						
aOR (95% CI)	1.59 (0.89; 2.85)	1.42 (0.84; 2.42)	1.28 (0.77; 2.13)	0.89 (0.46; 1.73)	0.8 (0.42; 1.52)	0.89 (0.49; 1.63)
**Android obesity**						
aOR (95% CI)	1.43 (0.80; 2.55)	1.35 (0.81; 2.28)	1.89 (1.11; 3.21) ^2^	0.94 (0.48; 1.84)	1.32 (0.68; 2.58)	1.39 (0.76; 2.61)

ORs (95% Cis) calculated by logistic regression and *p*-value corrected (Bonferroni). ^1^
*p* < 0.05; ^2^
*p* < 0.01, adjusted for socioeconomic status and Food consumption score; aOR: adjusted Odd Ratio.

**Table 4 nutrients-14-02465-t004:** Unadjusted differences in clinical and biological cardiometabolic markers of NCDs in the three SAM subgroups and unexposed controls.

	Marasmus			Kwashiorkor			Mixed-Type			Unexposed		
	N (Total)	%	Mean (SD)	N (Total)	%	Mean (SD)	N (Total)	%	Mean (SD)	N (Total)	%	Mean (SD)
**Anthropometry**												
Weight (kg)	141		53.5 (8.2)	167		55.7 (7.4)^2^	201		51.4 (7.5)^1,2^	396		55.1 (7.2)
Height (cm)	142		156.1 (8.8)	173		156.7 (9.1)	205		154.9 (9.1)^1^	406		157.6 (8.8)
Waist circumference (cm)	142		79.4 (9.1)	172		80.1 (9.2)^1^	205		78.0 (9.1)	406		77.9 (8.2)
Hip circumference (cm)	142		84.7 (8.9)	172		85.6 (8.4)^2^	203		83.3 (8.4) ^1,2^	405		86.0 (7.6)
Waist-to-Hip ratio (WHR)	142		0.94 (0.14) ^1^	171		0.93 (0.12)^1^	203		0.94 (0.11) ^1^	405		0.91 (0.11)
Waist-to-Height ratio WHtR	142		0.51 (0.06)	172		0.51 (0.06)^1^	205		0.50 (0.06)	406		0.49 (0.05)
Muscle strength (Kg)	106		30.7 (9.7)	122		30.1 (8.4)^1^	157		29.3 (8.0) ^1^	303		32.8 (8.8)
Body Mass Index (Kg/m^2^)	141		21.9 (2.9)	167		22.7 (2.8) ^2^	201		21.4 (2.7) ^1,2^	396		22.2 (2.5)
**Blood pressure (BP) mmHg**	105			125			156			301		
Systolic BP			119.2 (12.9)			120.1 (13.6)			117.0 (13.1)			119.6 (13.2)
Diastolic BP			70.4 (10.6)			71.6 (11.5)			70.6 (10.1)			71.6 (10.1)
Mean BP			86.7 (9.9)			87.7 (10.4)			86.1 (9.8)			87.5 (9.5)
Pulse pressure			48.8 (11.9)			48.5 (13.6)			46.4 (10.9)			47.9 (12.7)
**Glucose homeostasis**												
Fasting glycemia (mg/dL)	107		103.7 (17.1)	129		103.2 (14.5)	162		107.5 (17.3) ^1^	319		103.7 (14.5)
HbA1c (%)	30		4.6 (0.4) ^1^	30		4.7 (0.5) ^1^	30		4.6 (0.5) ^1^	52		4.1 (0.2)
**Lipids (mg/dL)**	118			134			172			331		
Total cholesterol			155.9 (35.8)			159.5 (34.6)			148.7 (35.5) ^1^			159.1 (36.6)
Non-HDL-C			112.5 (30.9)			113.4 (29.5)			106.3 (30.5) ^1^			114.6 (32.0)
HDL-C			43.4 (7.9)			46.1 (9.0) ^2^			42.3 (8.1) ^2^			44.4 (8.4)
LDL-C			92.0 (30.6)			93.6 (29.8)			86.2 (30.7) ^1^			94.2 (31.2)
TG ^3^			97.8 (74.6,128.3) ^3^			97.6 (74.5,127.6) ^3^			97.9 (75.1,128.9) ^3^			96.9 (74.7,126.4) ^3^
**Creatinine (mg/dL)**	117		0.88 (0.18)	133		0.86 (0.15)	171		0.87 (0.16)	331		0.88 (0.19)
**Albumin (mg/dL)**	118		4.4 (0.3)	134		4.4 (0.3)	172		4.3 (0.3) ^1^	328		4.4 (0.3)
**Thinness (BMI < 18.5)**	141	6.4		167	3.6		201	11.9 ^1^		396	3.8	

^1^ Significant difference from unexposed, ^2^ significant difference between mixed-type SAM and Kwashiorkor. ^3^ Geometric mean +/− SD. Means were compared using student’s *t*-test HDL-C = High-Density Lipoprotein Cholesterol. LDL-C = Low-Density Lipoprotein cholesterol TG = Triglycerides. HbA1c = Glycated Hemoglobin. BMI = Body Mass Index.

**Table 5 nutrients-14-02465-t005:** Adjusted difference (95% CIs) of clinical and biological cardiometabolic markers between groups.

Variable	Marasmus vs. Unexp	Kwash vs. Unexp	Mixed vs. Unexp	Kwash vs. Marasmus	Mixed-type vs. Marasmus	Mixed-type vs. Kawsh
BMI (kg/m^2^)	−0.04 (−0.79; 0.72)	0.50 (−0.19; 1.19)	−0.56 (−1.23; 0.10)	0.54 (−0.2; 1.40)	−0.53 (−1.36; 0.31)	−1.07 (−1.85; −0.28) ^2^
Weight (kg)	−1.25 (−3.2; 0.77)	0.91 (−0.94; 2.77)	−3.05 (−4.83; −1.26) ^3^	2.16 (−0.15; 4.48)	−1.79 (−4.05; 0.45)	−3.96 (−6.07; −1.85) ^3^
Height (cm)	−1.78 (−4.16; 0.59)	−0.52 (−2.68; 1.63)	−2.48 (−4.57; −0.4) ^1^	1.26 (−1.44; 3.97)	−0.7 (−3.34; 1.94)	−1.96 (−4.41; 0.48)
Waist circumference (cm)	1.85 (−0.56; 4.27)	1.99 (−0.19; 4.19)	0.15 (−1.96; 2.27)	0.14 (−2.61; 2.9)	−1.7 (−4.39; 0.98)	−1.84 (−4.34; 0.65)
Hip circumference (cm)	−1.38 (−3.62; 0.85)	−0.21 (−2.25; 1.81)	−2.27 (−4.24; −0.31) ^1^	1.16 (−1.39; 3.72)	−0.89 (−3.38; 1.59)	−2.05 (−4.37; 0.25)
Muscle strength (Kg)	−2.16 (−4.88; 0.56)	−2.35 (−4.88; 0.16)	−3.47 (−5.82; −1.11) ^2^	−0.19 (−3.34; 2.95)	−1.30 (−4.31; 1.7)	−1.11 (−3.94; 1.72)
Glycemia (mg/dL)	−0.53 (−5.56; 4.5)	−0.38 (−4.96; 4.19)	3.38 (0.92; 7.7) ^1^	0.14 (−5.63; 5.93)	3.92 (−1.64; 9.48)	3.77 (−1.38; 8.93)
HbA1c (%)	0.49 (0.18; 0.8) ^3^	0.59 (0.29; 0.88) ^3^	0.46 (0.17; 0.76) ^3^	0.09 (−0.25; 0.44)	−0.03 (−0.38; 0.32)	−0.12 (−0.45; 0.2)
total Cholesterol (mg/dl)	−2.62 (−13.56; 8.30)	0.28 (−9.84; 10.41)	−7.96 (−17.54; 1.61)	2.91 (−9.68; 15.5)	−5.33 (−17.48; 6.8)	−8.25 (−19.6; 3.19)
HDL-C (mg/dL)	−1.29 (−3.88; 1.3)	1.65 (−0.75; 4.05)	−1.86 (−4.14; 0.4)	2.94 (−0.05; 5.93)	−0.57 (−3.45; 2.3)	−3.51 (−6.23; −0.8) ^2^
Albumin (mg/dL)	−0.00 (−0.10; 0.09)	−0.02 (−0.11; 0.06)	−0.07 (−0.16; 0.01)	−0.02 (−0.13; 0.09)	−0.06 (−0.18; 0.04)	−0.04 (−0.15; 0.05)
Systolic pressure	−0.87 (−5.09; 3.34)	0.03 (−3.83; 3.9)	−2.07 (−5.72; 1.58)	0.91 (−3.93; 5.76)	−1.19 (−5.86; 3.47)	−2.10 (−6.46; 2.25)
Diastolic pressure	−1.27 (−4.54; 1.99)	−1.08 (−4.08; 1.91)	−0.77 (−3.6; 2.06)	0.19 (−3.56; 3.95)	0.49 (−3.12; 4.11)	0.3 (−3.07; 3.68)
Pulse Pressure	0.39 (−3.55; 4.34)	1.11 (−2.50; 4.73)	−1.29 (−4.71; 2.12)	0.72 (−3.81; 5.25)	−1.68 (−6.06; 2.68)	−2.41 (−6.49; 1.66)
Mean Pressure	−1.14 (−4.24; 1.95)	−0.71 (−3.55; 2.13)	−1.20 (−3.89; 1.47)	0.43 (−3.13; 3.99)	−0.06 (−3.49; 3.36)	−0.49 (−3.69; 2.7)
Non-HDL-C	−1.14 (−4.24; 1.95)	−0.71 (−3.55; 2.13)	−1.2 (−3.89; 1.47)	0.43 (−3.13; 3.99)	−0.06 (−3.49; 3.36)	−0.49 (−3.69; 2.7)
Creatinine	0.00 (−0.04; 0.05)	−0.01 (−0.06; 0.03)	0.00 (−0.04; 0.04)	−0.01 (−0.07; 0.04)	−0.00 (−0.06; 0.05)	0.01 (−0.04; 0.07)
TG (mg/dL)	1.01 (0.97; 1.04)	1.00 (0.96, 1.05)	1.01 (0.96, 1.05)	−0.21 (−0.71; 2.32)	0.01 (−0.04; 1.92)	−0.97 (−2.24; 3.05)
LDL-C (mg/dL)	−1.48 (−10.92; 7.96)	−0.55 (−9.31; 8.21)	−5.58 (−13.92; 2.75)	0.92 (−9.93; 11.79)	−4.10 (−14.60; 6.39)	−5.03 (−14.95; 4.89)
WHR	0.04 (0.00; 0.07) ^2^	0.02 (−0.00; 0.05)	0.02 (−0.00; 0.05)	−0.01 (−0.05; 0.02)	−0.01 (−0.05; 0.02)	0.00 (−0.03; 0.03)
WHtR	0.01 (0.00; 0.03) ^1^	0.01 (−0.00; 0.02)	0.00 (−0.00; 0.02)	−0.00 (−0.02; 0.01)	−0.00 (−0.02; 0.01)	−0.00 (−0.02; 0.01)

Adjusted difference (95% CIs) calculated by linear regression and *p*-value corrected (Bonferroni). ^1^
*p* < 0.05; ^2^
*p* < 0.01; ^3^
*p* < 0.001, Adjusted for socioeconomic status food consumption score.

## Data Availability

The data presented in this study are available upon request from the corresponding author. The data are not publicly available due to ethical requirements.

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
