# Peer review of "Risk of Chronic Disease after an Episode of Marasmus, Kwashiorkor or Mixed–Type Severe Acute Malnutrition in the Democratic Republic of Congo: The Lwiro Follow-Up Study"

_nutrients, 2022, doi:10.3390/nu14122465_

Round 1

Reviewer 1 Report

Dear authors, congratulations on your interesting manuscript. The introduction is consistent and describes the most important issues that introduce the reader to the topic. The material and methods are fully described. Study limitations also described in detail. I only have a few small suggestions. They are indicated in the manuscript in the form of comments.

Reviewer 2 Report

Anthropometric measurements were carried out at the time of the participant's accession to the project and in the subsequent period, follow-up? on what equipment were they performed? did the same person take all the measurements on the same equipment? please describe and with what accuracy? whether the children's weight and growth percentile grids for gender and age or BMI grids were used to assess nutritional status. 

Food consumption frequency - please describe how many questions it consisted of, which product groups it included, what was the consumption frequency scale. Was this questionnaire only for post-follow-up adults?

in general, the work is very interesting and deals with an important, little-known topic.
